# Comparative Study of Useful Compounds Extracted from *Lophanthus anisatus* by Green Extraction

**DOI:** 10.3390/molecules27227737

**Published:** 2022-11-10

**Authors:** Daniela-Simina Stefan, Mariana Popescu, Cristina-Mihaela Luntraru, Alexandru Suciu, Mihai Belcu, Lucia-Elena Ionescu, Mihaela Popescu, Petrica Iancu, Mircea Stefan

**Affiliations:** 1Faculty of Chemical Engineering and Biotechnologies, University Politehnica from Bucharest, No. 1-7, Gh. Polizu Street, Sector 1, 011061 Bucharest, Romania; 2Pharmacy Faculty, University Titu Maiorescu, No. 22, Dâmbovnicului Street, Sector 4, 040441 Bucharest, Romania; 3SC Hofigal Export Import SA, No. 2, Intrarea Serelor Street, Sector 4, 042124, Bucharest, Romania; 4National Military Medical Institute for Research and Development, No. 103, Splaiul Independenței, Sector 5, 050096 Bucharest, Romania

**Keywords:** *Agastache foeniculum* (*Lophanthus anisatus*), bio-alcoholic extraction, essential oil, supercritical fluids, antimicrobial effect

## Abstract

Essential oils were obtained from different parts of *Agastache foeniculum* (*Lophanthus anisatus*) plants by means of extraction: green extraction using hydro-distillation (HD) and bio-solvent distillation, BiAD, discontinuous distillation, and supercritical fluid extraction, in two stages: (1) with CO_2_, and (2) with CO_2_ and ethanol co-solvent. The extraction yields were determined. The yield values varied for different parts of the plant, as well as the method of extraction. Thus, they had the values of 0.62 ± 0.020 and 0.92 ± 0.015 g/100 g for the samples from the whole aerial plant, 0.75 ± 0.008 and 1.06 ± 0.005 g/100 g for the samples of leaves, and 1.22 ± 0.011 and 1.60 ± 0.049 g/100 g for the samples of flowers for HD and BiAD, respectively. The yield values for supercritical fluid extraction were of 0.94 ± 0.010 and 0.32 ± 0.007 g/100 g for the samples of whole aerial plant, 0.9 ± 0.010 and 1.14 ± 0.008 g/100 g for the samples of leaves, and 1.94 ± 0.030 and 0.57 ± 0.003 g/100 g for the samples of flowers, in the first and second stages, respectively. The main components of *Lophanthus anisatus* were identified as: estragon, limonene, eugenol, chavicol, benzaldehyde, and pentanol. The essential oil from *Agatache foeniculum* has antimicrobial effects against *Staphylococcus aureus*, the *Escherichia coli* and *Pseudomonas aeruginosa*. Acclimatization of *Lophantus anisatus* in Romania gives it special qualities by concentrating components such as: estragole over 93%, limonene over 8%, especially in flowers; and chavicol over 14%, estragole over 30%, eugenol and derivatives (methoxy eugenol, methyl eugenol, etc.) over 30% and phenyl ether alcohol over 20% in leaves. As a result of the research carried out, it was proven that *Lophanthus anisatus* can be used as a medicinal plant for many diseases, it can be used as a spice and preservative for various foods, etc.

## 1. Introduction

*Agastache foeniculum* (*Lophanthus anisatus*) is one of the aromatic plants in the genus *Agastache*—included in *Nepetoideae*, a subfamily of *Lamiaceae*. It is native to the USA and Canada, and it can also be found under common names such as “fennel giant hyssop”, “anise hyssop”, or “Mexican mint” [1].

It is a perennial, aromatic, medicinal, ornamental and melliferous plant, being a perspective for farmers. Therapeutically, it is used mainly for cardiovascular, nervous, and gastrointestinal disorders and for the treatment of colds, fever, and profuse sweating. It has anti-vomiting, antibacterial, and antifungal properties [2], reduces stomach acidity, and has a positive effect on people suffering from high blood pressure, angina and atherosclerosis [3,4]. Its infusion has a mild sedative effect and soothes and relieves headaches. It can also be used for phytotherapeutic treatments and can be successfully used in treating a wide range of respiratory conditions [5,6].

Recent experiments with extracts of anise hyssop leaves proved that they can be useful in anti-aging pharmacology [7]. Ownagh et al. studied the antifungal effects of essential oils (EO) extracted from Agastache, Thyme and Satureja, on *Aspergillus fumigatus*, *Aspergillus flavus* and *Fusarium solani*. The results showed that *Agastache* EO is effective as a fungistatic at concentrations of 2000 μL/mL [8].

It is listed in the top four melliferous plants in the world by American specialists. The flower blooms for a long time period, about 5–6 months [9].

Because of its genetic capacity to adapt to environmental conditions, it is cultivated and known worldwide. Several acclimatization studies of *Agastache foeniculum* (in Scotland [10], Ukraine [11], Belarus [12] Lithuania [13,14], Bulgaria [15,16], Finland [17], Iran [18], India [19], and Romania [20] have been presented in the literature.

The most valuable component of EO extracted from this plant is estragole, or methyl chavicol, which gives the flavor of *Agastache sp*. to the EO.

Transfer to areas with different climates can, however, greatly alter both the yields and the composition of the volatile oils produced by the plant, because the environmental factors can change the metabolic processes.

In Ukraine, the main components of the EO obtained from *Lophanthus anisatus* were pulegone (60.04%) and -isomenthone (12.59%) [21]. It was also found that *Agastache* is a source of chemical elements (potassium and phosphorous) involved in human body metabolism [11].

For *Lophanthus anisatum Benth*, introduced in the Astrakhan region, Yurtaeva et al. found as main components luteolin (47.80%), rutin (2.57%), and quercetin (3.61%) [22]. Luteolin is a flavonoid (polyphenolic secondary plant metabolite) that was used as natural yellow dye, but now it is valorized for its activity against hypertension, inflammation, neurological disorders, and cancer [3], whilst rutin is a flavonoid with antioxidative, antihypertensive, antidiabetic, anti-inflammatory, and cardioprotective activities [4].

The oil yields depend on a series of factors, such as: the harvest time, the irrigation regime, the addition of fertilizers and the sowing time, the method of preliminary drying of the plants and the plant density upon the cultivated surface.

The highest oil yield extracted from *A. foeniculum* was in the middle of the blooming period [23]. The effect of the harvest period was also studied by Duda et al. in 2015 [24]. The authors reported that the best moment for harvesting the plant is the beginning of blooming time, and in the afternoon.

Concerning the irrigation regime (severe water deficit being favorable), it was found that the highest amount of EO (2.30 %, vegetative stage) was extracted from plants that were irrigated at 55% of field water capacity (severe water deficit), and the lowest one (1.64 %, vegetative stage) was obtained from plants that received between 70 (moderate drought stress) and 100% (no drought stress) of field water capacity [25,26].

The effect of organic and inorganic fertilizers on content and constituents was studied by Omidbaigi [27]. It was showed that nitrogen accessibility had a significant effect on the EO percentage extracted from plants: 2.88% EO was extracted from plants treated with 100 kg/ha nitrogen, while the lowest EO content (2.1 and 2.3%) was obtained from plants that received 50 kg/ha nitrogen and 0 kg/ha nitrogen, respectively.

While the percentage of EO extracted was influenced by the nitrogen addition, the composition of EO appeared not to be influenced by the amount of added nitrogen (with the main component, estragole, of about 95%). Additionally, the EO composition of *Agastache foeniculum* (*Lophanthus anisatus*) cultivated in Iran was presented with the main component of extracted EO being estragole, in a proportion of 87.5% [28]. Omidbaigi also indicated the effect of sowing time [29]. The authors showed that the sowing time influences the composition of EO extracted, with the content in estragole decreasing from the end of March (at the sowing time 20th March (of)—of 92.12%) to the end of June (at the sowing time 20th June (of)—45.6%). Contrary to the estragole content, the percentage of limonene increased from 5.92% to 48.8%, for the two sowing dates. In the study by Moghaddam et al. (2015), several types of fertilizers were used, namely nitrogen fertilizer (50 kg ha^−1^), vermicompost (30 t ha^−1^), cow manure (20 t ha^−1^), cow manure (25 t ha^−1^), and a combination of vermicompost and cow manure (30 t ha^−1^ + 20 t ha^−1^). The results showed that organic manure improved the growth and yield of the *Agastache foeniculum* crop [30].

The oil yields were significantly affected by the drying procedure. When drying was performed in an oven at 60 °C, all of the oil was lost, so that no parameters could be further measured and compared. The plants dried at room temperature (25 °C) showed the highest EO content (2.2%), whereas a lower percentage was obtained from those plants dried in an oven at 40 °C (1.6%) [31]. The shares of oil from different parts of *A. foeniculum* dry plant were found to be: leaves—35.9%; inflorescences—19.0%; heartwood—25.8%; strain—19.3% [20].

In Romania, the production of herbs (dry weight) was between 3.05 and 3.83 t/ha, and was inversely proportional with the plant density, for the three densities applied (47,619 plants/ha 35,714 plants/ha and 28,517 plants/ha) [20].

The composition of volatile oils obtained from plants belonging to the *Lamiaceae* family, as well as their antimicrobial activity, was presented in Shutava [32] and Karpiński [33]. The composition of the volatile oils obtained from *Agastache foeniculum* has been studied in numerous articles dating as far back as 1945 (Polak) [34], and even further. In the EO extracted from *Agastache foeniculum*, over 50 compounds were identified and isolated. Only 10 constituents, however, accounted for more than 0.1%. Prevailing as the major constituents were the methyl chavicol or estragole—up to 97% [35], and limonene—up to (3.6–3.9%) [19], 1,8-cineole (2.0%), and globulol (1.4%) [28]. Other components of the EO were: monoterpenes, sesquiterpenes, oxygenated monoterpenes, oxygenated sesquiterpenes, camphene, myrcene, and phenolics, among others. Over 50 compounds were detected in EO, and headspace of 14 different clones of *Agastache foeniculum* obtained by hydro-distillation and by purging the plant material with nitrogen gas, respectively [36]. The main component was methyl chavicol, which is also known as estragole, the content in dry plant being in the range of 94–97%.

Some differences between the composition of oils obtained from leaves and those obtained from flowers have been reported by Charles (1991) [37]. Differentiation of carotenoid content on different parts of the plant was presented by Chae (2013) [38]. The content of *Phenylpropanoids* (mainly rosmarinic acid, tilianin, and acacetin, in different parts of the plant *Agastache foeniculum*, and *Agastache foeniculum* ‘Golden Jubilee’ was presented by Woo Tae Park (2014) [39]. Rosmarinic acid accumulation was higher in the roots of both species, compared to the flower, leaf and stem. Tilianin accumulation was higher in all ‘Golden Jubilee’ organs, compared to *A. foeniculum*. Acacetin accumulation was lower in the various plant parts of both plants, the greatest amount of acacetin being accumulated in the flower.

The phytochemistry of aromatic and medicinal plants from the genus *Agastache* has been presented in detail by many authors [10,17,19,23,39,40,41,42,43,44,45,46,47,48,49,50]. It was shown that molecules such as estragole, 1,8-cineole, terpineol-4, and g-terpinene have antifungal activity against *Trichophyton erinacei*, *T. menta grophytes*; *T. rubrum*; *T. schoenleinii*; and *T. soudanense* [40,48]. Some research has shown EO from *Agastache foeniculum* as presenting antifungal, antioxidant [18,41], anticancer [35], or insecticide [42] activities. The rosmarinic acid contained in large quantities in these EOs has been shown to have anticancer properties [6]. The chemical and antifungal properties of the EO essential oil from *Agastache foeniculum* were also presented by Kutchin et al. (2017) [5]. It was found that the EO of *Lophantus anisatus* has high antifungal activity [18].

Estragole and limonene are included in the majority of EOs already in use, or proposed for antimicrobial [43,44,45] or antifungal [2] products. Recently, EOs have been also suggested as valuable components for biopesticides [46], or as alternative antimicrobial food preservatives [47,48,49,50].

For essential oils (EOs) and other useful compounds/extracts from different parts of the aromatic plant, extraction can be applied through traditional approaches: steam and hydro-distillation, liquid-solvent extraction and distillation; and by modern methods: microwave-assisted extraction, supercritical fluid, and subcritical water extractions [51,52]. A new extract concept is green extraction, which uses alternative solvents and principally water or agro- or bio-solvents, diminishing the consumption of petrochemical solvents and volatile organic compounds (VOCs) that are flammable, volatile, and often toxic, being responsible for environmental pollution and the greenhouse effect. The agro- or bio-solvents represent a renewable resource produced from biomasses such as: wood, starch, vegetable oils, or fruits, having good solubilizing power, and being biodegradable, non-toxic and non-flammable [53].

Supercritical CO_2_ extraction (SFE) is another green technique with many advantages over conventional methods in terms of toxicity, selectivity, and avoidance of compound degradation, being a green process as well [54,55,56].

The aim of this study was to identify valuable compounds and to determine their compositions in extracts obtained from different parts of plants from the species *Lophantus anisatus*) acclimatized in Romania, at the Vegetable Research and Development Station (VRDS) Buzau. Three green extraction methods were evaluated: a) discontinuous distillation methods such as hydro-distillation (HD) and bio-solvent (alcoholic from plums) distillation (BiAD), and b) continuous extraction methods such as supercritical fluid extraction with CO_2_ and with CO_2_ and ethanol as co-solvent. The quality of extracts was evaluated by the antimicrobial action of the extracts on selected pathogenic bacteria.

*Agastache foeniculum* (*Lophantus anisatus*) plants, used in the studies carried out, were harvested in the middle of the blooming period, from a plot of land with moderate drought stress, fertilized with organic fertilizers. The plant density was around 40,000 plants/ha.

## 2. Results

### 2.1. Green Oil Extraction Yields

#### 2.1.1. HD and BiAD Oil Extraction Yields

Experiments were developed using aerial parts of the plants of *Lophanthus anisatus*. The composition (in dried mass percentage) of the aerial part of the plants was: leaves—27.3%; flowers—36.4%; strain—36.3%.

Comparing with the data from the literature, [20], it can be observed that the plants acclimatized at VRDS Buzau, Romania had a greater weight in terms of the inflorescence (flower) compared to the weight of the leaves, which represented a gain in terms of the amount of useful compounds that could be extracted, flowers being proven to be the main sources of volatile oils.

Volatile oils were separated from the distillates obtained from the whole aerial part, flowers and leaves of *Lophantus anisatus* by hydro extraction (HD) and bio-alcohol extraction (BiAD). The oil extraction yields obtained by HD and BiAD are presented in Table 1.

The extraction yields varied between 0.6 and 1.3 g/100 g dried plant for HD and between 1 and 1.6 g/100 g dried plant for BiAD.

The best 1.6% yield of volatile oil extraction was obtained by extraction with BiAD from flowers. The *Lophanthus anisatus* plants subjected to analysis were grown in areas with moderate drought stress. The yield values were comparable to those obtained in the literature, [28,29] under the same growing conditions. In order to increase the yield of volatile oil extraction, it is recommended to reduce the water regime used during the growing season.

#### 2.1.2. SFE Extraction

##### Supercritical Extraction Process Yield

The samples and extracts of *Lophanthus anisatus* obtained using SFE are presented in Table 2. Three types of samples (flowers, leaves and the whole plant) were subjected to two stages of SFE. In the first stage, active compounds were extracted with supercritical CO_2_ as solvent, and in the second stage, 10% wt% ethanol was added to the exhausted sample from the first stage to enhance the extraction efficiency. Strong smelling oily extracts were obtained, and the efficiency after 8 h of extraction (first stage) varied between 0.90 ± 0.010 g extract/100 g leaf sample and 1.94 ± 0.030 g extract/100 g flower sample. In the second stage, 23–34% more extract was obtained, as presented in Table 2. Extraction yields of volatile oils obtained by SFE were much improved compared to those obtained with classic extraction methods, reaching 3.04% for leaves and 2.51% for flowers.

In Figure 1a, the extraction curves for all types of samples in the first extraction stage are presented. These curves are of the same shapes as those depicting the classical behavior of the CO_2_ supercritical extraction process: the first period extraction rate depends on the solubility of compounds in solvent, and in the second period, the extraction rate is controlled by phase diffusion. From the flower samples (red points), 2.9 g of extract was obtained in the first 320 min, then the extraction curve began to be flatten, and the process was no longer reliable. A three times smaller amount of extract was obtained, under the same conditions, from the leaf samples (green points), and from the whole plant (blue points). In the second extraction stage (Figure 1b), the exhausted samples were mixed with ethanol as co-solvent, and, due to the improving of solvent polarity, more extract was obtained, especially from the leaf samples (1.7 g extract—stage 2).

For the extraction process, the solvent consumption rate influences the process efficiency. For the supercritical extraction process, CO_2_ recycling leads to smaller consumptions. The extraction yield vs. CO_2_ consumption is presented in Figure 2, for both stages. The highest extraction yield of 1.94% was obtained from flower samples, for a CO_2_ consumption rate of 400 g/g of extract, in stage 1 (Figure 2a). In the second stage (by adding co-solvent), this consumption rate decreased to 300 g/g of extract (Figure 2b). The CO_2_ make-up per batch was 3 kg.

##### SFE Extraction Mass Balances

Supercritical extraction yields were calculated for all types of samples based on mass balances, as presented in Figure 3. From 150 g samples, 3.76 g extract were obtained from flowers, 3.05 g extract from leaves, and 1.89 g extract from whole plant.

##### SFE Extract Qualitative Analysis

From the analysis of the extract samples, the existence of two phases was observed: oil and solid, with the largest amount of oil being extracted from flowers. In Figure 4a, the oil content in all extracts obtained in the first extraction stage is presented (by weighing of extract phases): 33% oil present in extracts from flowers, and 50% oil from whole plant extract. The extract obtained from the second stage contains traces of ethanol (Figure 4b).

### 2.2. Total Polyphenolic Active Compounds Obtained by Green Extraction

The content of polyphenols is important for the antioxidant activity against radicals and reactive oxygen species [46]. The concentrations of total polyphenolic active compounds expressed in Gallic acid equivalent resulting from all used extraction methods (HD BiAD and SFE), as determined by the UV-VIS spectrometry method, are shown in Table 3. The largest amount of polyphenols was present in *Lophanthus anisatus* leaves, followed by flowers, and the whole plant (without roots), a fact confirmed by all three types of extractions. In the literature, strong differences in composition have been reported between the different parts of *Lophanthus anisatus* plant [38,39]. The use of the SFE method favors the extraction of polyphenols, followed by bio alcohol extraction and extraction with distilled water.

One can observe that the samples subjected to SFE using only supercritical CO_2_ as solvent showed values between 9.75–14 mg/g dried plant, In the second stage of SFE, ethanol was added to improve the polyphenol recovery, from 14 to 20 mg/g dried plant for the whole plant, flowers and leaves, respectively. The Gallic acid equivalent values were between 4.25 and 9.5 mg/g for the samples obtained by HD, and between 8.5 and 12.25 mg/g for extraction obtained by BiAD, for the whole plant and leaves, respectively.

#### Polyphenolic Compound Concentrations in the Extracts Obtained by HD, BiAD and SFE Determined Using the HPLC Method

Using the HPLC method, we determined the concentrations of polyphenols frequently mentioned in the bibliographic sources, namely: chlorogenic acid, caffeic acid, rosmarinic acid apigenin, 7-glucoside, and ferulic acid [12,19]. The percentage of these polyphenols in total polyphenol varied between 30–38% in extracts from whole plant (P1, P2, P11, P12), 27–35% in extracts from flowers (P5–P8), and 26–31% in extracts from leaves (P3, P4, P9, P10).

From the results presented in Table 4 regarding the composition in various polyphenols determined by high performance liquid chromatography (HPLC), higher concentrations could be observed in rosmarinic acid (a phytochemical compound with real anticancer properties), as reported by Woo Tae Park (2014) [39]. The samples obtained in the first stage of extraction with supercritical CO_2_ contained rosmarinic acid between 227 and 279 mg/100 g of dried plant. Adding ethanol in the second stage of SFE, the amount of rosmarinic acid increased by 128–138 mg/100 g of dried plant. The samples subjected to extraction by bio-alcohol distillation (BiAD) showed rosmarinic acid values of 206 mg, 80 mg, and 30.4 mg/100 g for the leaves, flowers, and the whole plant, respectively. The lowest values for rosmarinic acid in the samples obtained by hydro-distillation (HD) were as follows: 120.6 mg, 204 mg, and 191.1 mg/100 g for whole plant, leaves, and flowers, respectively.

Chlorogenic acid and caffeic acid were present in all of the samples studied (Table 4): flowers, leaves, and the aerial part of the plant. Most of them were extracted with SFE from leaves: 41.7 mg and 21.3 mg chlorogenic acid/100 g and 14.2 mg and 12.8 mg caffeic acid/100 g, in the first and second stages of extraction, respectively.

The other compound determined by HPLC was ferulic acid, which was found in higher amounts in bio-alcoholic extracts from *Lophanthus anisatus* flowers (3.15 mg/100 g of dry plant) and in the SFE extract from whole plant (stem) obtained using supercritical CO_2_ and ethanol (13.5 mg/100 g of dry plant). Apigenin 7-glucoside was observed to accumulate in higher amounts in leaves.

### 2.3. The Composition of Other Active Compounds Obtained by Green Extraction

The extracts obtained using green methods (HD, BiAD and SFE) were analyzed using GC-MS to identify the composition of extracted compounds. Values for P1–P6 extracts obtained by HD and BiAD are shown in Table 5, while those for SFE extracts (P7–P12) are listed in Table 6.

Of the 30 chemical compounds present in volatile *Lophanthus anisatus* oil, estragole and limonene (compounds with proven antifungal and antimicrobial effects) were present in most extracts obtained by the three methods (HD, BiAD and SFE). Estragole was found in higher quantities in the aerial part of the plant, for all types of extracts, for HD and BiAD (in a range from 66 to 90 %), while the values of SFE extracts were between 80% and 93%. Estragole concentration rose to over 90% by extraction with BiAD and between 92% and 93% by extraction with SFE in flower extracts.

In the extracts obtained by HD and BiAD from leaves, estragole was found in smaller quantities, its concentration being in the range of 18–30%. Other compounds were highlighted, such as havicole (around 14%), eugenol (between 13 and 18%), benzaldehyde (between 2 and 11%), pentanol (between 3 and 9%), benzyl alcohol (between 2 and 4%) and phenyl ethyl alcohol (in proportion of 20%). In the SFE extracts (see Table 6), these components were less highlighted. Limonene was present in greater quantities (8%) in the extracts obtained from *Lophanthus anisatus* flowers, regardless the extraction method, while in the leaves and whole plant, the values were smaller. Concentrations of limonene found in *Lophanthus anisatus* acclimatized to Buzau were higher than the range mentioned in the literature of 3.6–3.9%, [19] but the 1,8 cineole and globulot were not identified.

Eugenol, the compound with antiseptic and anesthesia properties, was identified in higher percentages in the extracts obtained by HD, BiAD, and SFE from *Lophanthus anisatus* leaves, with values of 13, 18 and over 4 %, respectively.

Eugenol was not mentioned in the literature as a main component *of Lophanthus anisatus.* It seems that it is found in leaves, especially in the species acclimatized in Romania.

### 2.4. Antimicrobial Activity of Essential Oils from Lophanthus anisatus

A comparison of the antimicrobial activity of essential oils resulting from experiments and mint extract (leaves and aerial parts) is presented in Table 7.

Antimicrobial activity of essential oils was evaluated using the conventional semi-quantitative interpretation of inhibition zone: resistant—if is no inhibition area; intermediate—if the diameter of the inhibition zone is below 20 mm; and sensitive—if the diameter of the inhibition zone is over 20 mm.

From the analysis of Table 7, it can be seen that EO from *Lophanthus anisatus* flowers had intermediate antimicrobial activity against *Staphylococcus aureus*. The flower infusion had intermediate antimicrobial action against *Escherichia coli* and *Pseudomonas aeruginosa*. *Lophanthus anisatus* had a better antimicrobial action than mint leaf infusion or decoction.

Table 8 lists the antibiofilm effects of *Lophanthus anisatus* against Staphylococcus aureus, *Escherichia coli* and *Pseudomonas aeruginosa*.

After evaluation of the antibiofilm effect, EO from *Lophantus anisatus* was found to be active only against *Staphylococcus aureus.*

## 3. Discussion

The acclimatization of *Lophanthus anisatus* in Romania can be considered as being successfully performed. The relatively good yield of obtained essential oils, the chemical composition and the contents of total polyphenolic active compounds were in accordance with the data for this plant cultivated in other countries. The composition (in dried mass percentage) of the aerial part of the plants was: leaves—27.3%; flowers—36.4%; whole plant heartwood—36.4%.

The extraction yields for HD varied between 0.6 to 1.3, and between 1 to 1.6 g/100 g dried plant with BiAD, less than the yields obtained with the SFE method.

Experimental studies on supercritical extraction of *Lophanthus anisatus* samples (flowers, leaves and whole plant) showed that this method can be used as a viable technology to extract volatile and nonvolatile compounds such as polyphenols and other compounds, with yields of 2–3 g/100 g of solid sample. This yields can be improved by 25–50%, by adding ethanol as co-solvent. Supercritical extraction process operating conditions of 40 MPa pressure and temperature of 40°C, and extraction time of 400 min in the first extraction stage and 200 min in second extraction stage (by adding co-solvent) can be chosen to obtain 2–3 g oily extract from flowers of *Lophanthus anisatus.*

The extracts can contain up to 20 mg/g total polyphenols. The largest amount of polyphenols is present in *Lophanthus anisatus* leaves, followed by flowers and the whole plant (without roots), a fact confirmed for all three types of extractions. The use of supercritical CO_2_ favors the extraction of polyphenols, followed by BiAD and HD. The main polyphenol present in *Lophanthus anisatus* is rosmarinic acid, with concentrations of around 270 mg/100 g of dried plant. Over 30 chemical compounds were identified in the whole aerial *Lophanthus anisatus* plant, and the composition depended on the extraction type and on the component parts of the plant.

The major components of the EO extracted from the *Lophantus anisatus* plants are, estragole in range of 60–93%, eugenol, methyl eugenol, limonene, retinol etc., their proportion depending on the extraction method. In the case of sample P1, the major component of EO is the retinol, which is vitamin A1, and is mainly used as dietary supplement. Sample P3 contains mainly estragole and eugenol in the same proportion Estragole is one of the natural present phenylpropanoids, having antifungal and antioxidant properties [57]. Eugenol is used in medical applications (as antiseptic, anaesthetic, in dentistry) [58,59]. The highest percent in limonene was obtained for the P4 sample. Limonene is well known for its antinociceptive properties (increases tolerances or reduces sensitivity to a dangerous or harmful stimuli) [60].

Tests on the antimicrobial activity of essential oils showed that they are active against *Staphylococcus aureus*. Therefore, they can present interest as alternative antimicrobial food preservatives, instead of the synthetically produced food additives.

Acclimatization of *Lophanthus anisatus* in Romania gives it special qualities by concentrating components such as: estragole over 93%, limonene over 8%, especially in flowers; and chavicol over 14%, estragole over 30%, eugenol and derivatives (methoxy eugenol, methyl eugenol, etc.) over 30% and phenyl ether alcohol over 20% in leaves.

As a result of the research carried out, it was proven that *Lophanthus anisatus* can be used as a medicinal plant for many diseases, as a spice, as a preservative for various foods, etc.

## 4. Materials and Methods

### 4.1. Sample Preparation

The main objectives of the Laboratory of Genetics Breeding and Biodiversity Conservation from VRDS Buzau are obtaining competitive new biological creations, as required by growers and consumers, rehabilitating neglected plants in culture, acclimatizing new species, and promoting their culture.

For conducting the experiments, dried *Lophanthus anisatus* plants were used. The aerial parts of plants were dried slowly, at room temperature, away from the sun. Extraction experiments using the entire aerial part and its components (leaves, flowers and heartwood) were conducted after these parts had been chopped by cutting and then ground.

### 4.2. Extraction Methods

#### 4.2.1. Obtaining the Essential Oil and Extracts by the Green Methods: Hydro-distillation and Bio-Alcoholic Solvent Distillation Methods

For obtaining the essential oils and extracts, hydro-distillation (HD) and bio-alcoholic solvent distillation (BiAD) were used. The essential oils and extracts were evaporated by heating a mixture of water or bio solvent and plant materials, followed by the liquefaction of the vapors in a condenser at 100 °C and 78.37 °C, respectively. In a distillation vessel with a capacity of 17 L were placed, in turn, 2 kg of whole ground plant, ground leaves and ground flowers and 10 L of liquid, distilled water or bio-alcohol of 35 alcoholic degrees, respectively. Three liters of extract were collected at each extraction. Details of the working method and extraction yields are presented in Table 9.

To prevent the mixture from sticking to the bottom of the container, as well as the loss of material through the upper part, we used stainless steel grills. The setup used for the essential oil extraction by discontinuous distillation is presented in Figure 5.

The stability of the installation was ensured by means of two stands. A 2000 g amount of dried *Lophanthus anisatus* plant (the whole aerial plant, and part of the plant: leaves and flowers, respectively) were introduced into the vessel, 3, with a capacity of 17 L, and covered with 10 L of distilled water (HD) mixed with 35 % bio-alcohol (BiAD), respectively. The mixture was heated by an electric hob, 2. The temperature inside the boiling vessel was monitored by means of a thermometer, 4. The extraction vessel was connected to the cooling refrigerant, 6, through a steam collection pipe, 5. The refrigerant had two enclosures, the outer jacket through which the cold water circulated, with a role in cooling the collected vapors, and the inner channel through which the condensate circulated. The latter was collected in a graduated cylinder, 7. The obtained products were analyzed to determine the chemical composition by various methods and used for antimicrobial activity trials.

#### 4.2.2. Supercritical CO_2_ Extraction, SFE

Three kinds of solid samples were prepared for supercritical extraction: flowers, leaves, and whole plant. The dried samples were ground using a grinder (Tarrington House, KM150S) to reduce the particle size and to increase the extraction efficiency. For an extraction batch, a solid sample of 150 g was placed into the extraction vessel. For supercritical extraction, we used CO_2_ with 99.9% purity from Linde Gaz Romania as solvent, with ethanol of 99.8% purity from Sigma Aldrich (Germany), as co-solvent. Supercritical CO_2_ extractions were carried out in a high-pressure extraction unit laboratory pilot plant (HPE-CC 500, Eurotechnica GmbH).

The plant was equipped with a 3.2 L extraction vessel, a pump with a maximum flow rate of 30 L/h, a CO_2_ buffer tank, and one separator. The pressure in the extractor was maintained constant with a back pressure regulator, and the temperature of CO_2_ flow was achieved using heat exchangers. The pressure and the CO_2_ flow rate were maintained constant in the extractor, using the pump stroke and a back-pressure regulator, with the flow rate being measured with a gas flow meter. CO_2_ was recycled inside the plant, taking different states, from cooled liquid to gas–liquid mixture, and (in) supercritical conditions, to assure the density and solubility needed for the extraction of components from the solid samples. Samples were introduced into the extractor; extraction time and operating conditions were set up, and the extract was collected in a separator. Extract samples were taken every 30 min. The experimental conditions for volatile and nonvolatile compounds in supercritical extractions were varied as follows: extraction pressure, 100–400 bar; extraction temperature, 40 °C; CO_2_ flow rate, 13 kg/h. A flow diagram of the supercritical extraction process is presented in Figure 6.

Solid samples were dried and ground, and then introduced into the extractor. CO_2_ was heated and compressed to achieve supercritical conditions in the extractor. The extract and CO_2_ were expanded and separated into the separator. The extracts were collected, and the CO_2_ was collected and recycled into the process. The extraction process was followed by a cleaning process with ethanol, and traces of the extract were collected and stored. To enhance the extraction efficiency, the supercritical extraction was performed in two stages: stage 1—8 h extraction from solid samples, and stage 2—4 h extraction from partially exhausted solid sample sprayed with 10 wt% ethanol (as a co-solvent). The extraction yield was calculated as the mass of extract (g) divided by 100 g of plant material fed into the extractor. The codes for the extracts obtained with supercritical fluids for the dried fractions of the whole plant, leaves, and flowers are presented in Table 10.

The obtained products were analyzed in order to determine their chemical composition by various methods and used for antimicrobial activity trials.

### 4.3. Physicochemical Determinations

#### 4.3.1. Determination of Total Polyphenolic Compounds by UV-VIS Spectrometry

Total polyphenolic active compounds were determined using the UV-VIS spectrometry method. The equipment used was a Jasco UV-VIS V-530 spectrophotometer. The analysis method followed the procedure from ISO 14502-1:2005 (E). Determination of substances characteristic of green and black tea—Part 1: Content of total polyphenols in tea—Colorimetric method using Folin–Ciocalteu reagent. The main chemicals used were reagent Folin–Ciocalteu (VWR BDH Chemicals)—dilution 1:10, and sodium carbonate (Merck)—solution 7.5%. The reaction resulted in a blue complex, whose absorbance/concentration was read on the calibration curve (gallic acid Sigma-Aldrich standard) at a wavelength of 765 nm.

#### 4.3.2. Determination of Polyphenolic Compounds by High Performance Liquid Chromatography (HPLC)

In order to determine the composition of the samples obtained by CO2 supercritical fluid extraction, a Hitachi Chromaster HPLC system was used, equipped with a 5160 pump, 5310 column oven, 5260 thermostat autosampler, and a 5430 DAD detector. The separation was performed on a ZORBAX SB-C18 4.6 × 150 mm, 3.5 µm column. An adapted RP-HPLC method was developed. The mobile phase was a mixture of acetonitrile—methanol 1:1 *v*/*v* with 1% formic acid (A) and water with 1% formic acid (B), the elution being gradient at 1 mL/min as follows: 0′: 10% A—90% B; 5′: 30% A—70% B; 20′: 40% A—60% B; 25′: 42.5% A—57.5% B; 26′: 10% A—90% B; 30′: 10% A—90% B. Standard stock solutions were prepared by dissolving reference standards in methanol, and individual concentrations were in the range of 11.7 µg/mL–193 µg/mL. If needed, some samples were diluted with methanol prior to HPLC injection. All samples and reference standard solutions were filtered through a 0.2 µm PTFE filter, and 5 µL of each solution was injected into the HPLC system. Data acquisition was performed at 320, 285, 267 and 369 nm.

#### 4.3.3. Mass Spectroscopy

The extracted compounds were further analyzed by GC-MS. The analyses were undertaken with the help of a Thermo Electron Corporation Focus GC gas chromatograph, with a Macrogol 20,000 R column (film thickness of 0.25 μm, length of 60 m, and diameter of 0.25 mm). One milliliter of sample was diluted by adding 9 mL hexan *R* and filtered through a 0.22 µm sieve. The mobile phase used was helium at a flow rate of 1.5 mL/min, while the sample injection volume was 1.0 μL. A Thermo Electron Corporation DSQII mass spectrometer was used for detection. Identification of components in the samples analyzed by gas chromatography was carried out by comparing the sampled spectral peaks with spectra from a Wiley database. The working conditions are presented in Table 11.

### 4.4. Antimicrobial Activity of Essential Oils from Lophanthus anisatus

The antimicrobial activity against representative bacteria in the current pathology (*Staphylococcus aureus*, *Escherichia coli* and *Pseudomonas aeruginosa*) was performed by adapting the working method to efficiency of the results, costs and appropriate accuracy.

The experiments were performed using adapted standard Kirby–Bauer disk diffusion technique. The Mueller–Hinton agar medium was seeded with each bacterial suspension adjusted at 0.5 McFarland density (1.5 × 10^8^ CFU/mL) for each pathogenic bacterial representative of the hospital flora, namely *Staphylococcus aureus*, *Escherichia coli*, and *Pseudomonas aeruginosa.* The handling of the infectious material was performed under a BSC-EN class II biosafety hood. The substances were placed multiple times on the surfaces of the media as follows:—10 µL on discs with a diameter of 6 mm; 25 µL on discs with a diameter of 9 mm; 100 µL in glass cylinders with a diameter of 6 mm.

The tested substances, together with the appropriate controls and blanks, were aerobically incubated at 37 °C in a Memmert incubator for 24 h. The antimicrobial agent diffused into the agar and inhibited germination and growth of the tested microorganism. Reading was performed by examining the obtained inhibition zones measured in two different diameters (minimum and maximum). The arithmetic means per test and the arithmetic mean on the type of extract were computed.

The screening analysis of the possible effect against the bacterial biofilm was performed by the ring technique, i.e., by placing rings with a diameter of 3 cm on the surface of the Mueller–Hinton agar medium seeded with the bacterial strains introduced in the study. A 1000 µL amount of mother solution substance was pipetted, and the plate was incubated for 24 h at 37 °C to form a biofilm [61].

## Figures and Tables

**Figure 1 molecules-27-07737-f001:**
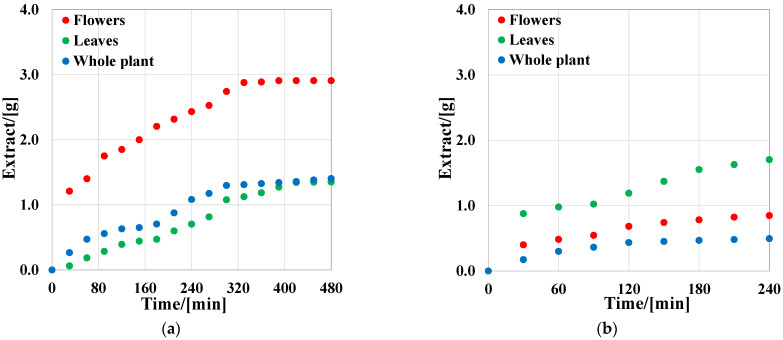
Extraction curves for plant samples. (**a**) First extraction stage: solid sample; (**b**) second extraction stage: solid sample + 10 wt% co-solvent.

**Figure 2 molecules-27-07737-f002:**
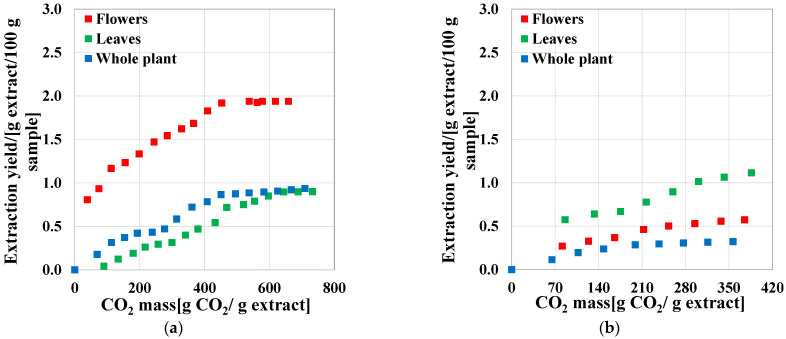
Extraction yields obtained from plant samples (**a**) First scenario: solid sample; (**b**) second scenario: solid sample + 10 wt% co-solvent.

**Figure 3 molecules-27-07737-f003:**
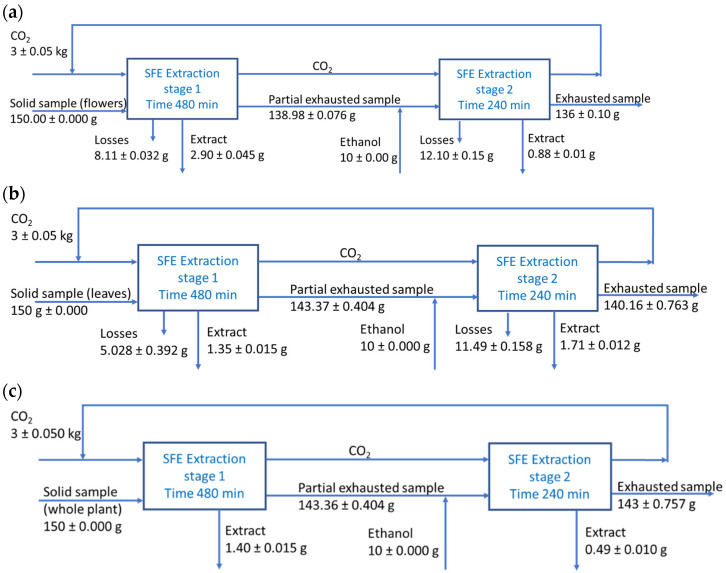
Mass balances for supercritical extraction of *Lophanthus anisatus* samples: (**a**) flowers; (**b**) leaves; (**c**) whole aerial plant.

**Figure 4 molecules-27-07737-f004:**
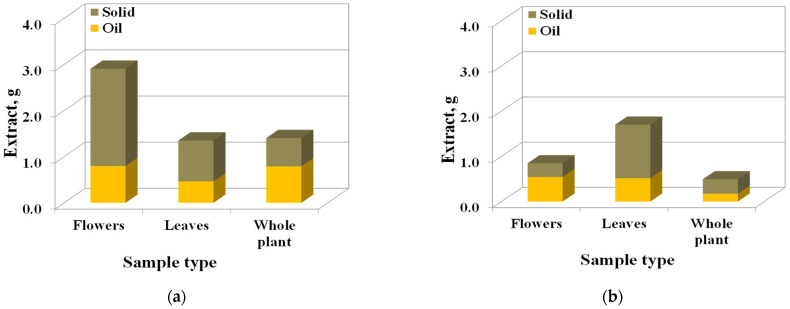
Total extract oil/solid ratios: (**a**) first extraction stage; (**b**) second extraction stage.

**Figure 5 molecules-27-07737-f005:**
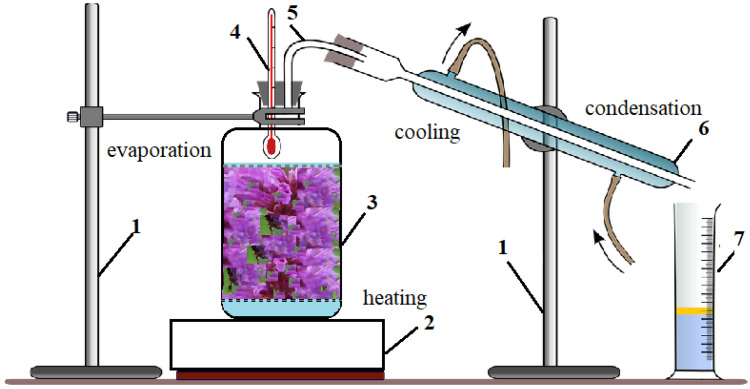
Schematic diagram of discontinuous distillation method (1—stands, 2—electric hob, 3—extraction vessel, 4—thermometer, 5—collection pipe, 6—refrigerant, 7—graduated cylinder).

**Figure 6 molecules-27-07737-f006:**
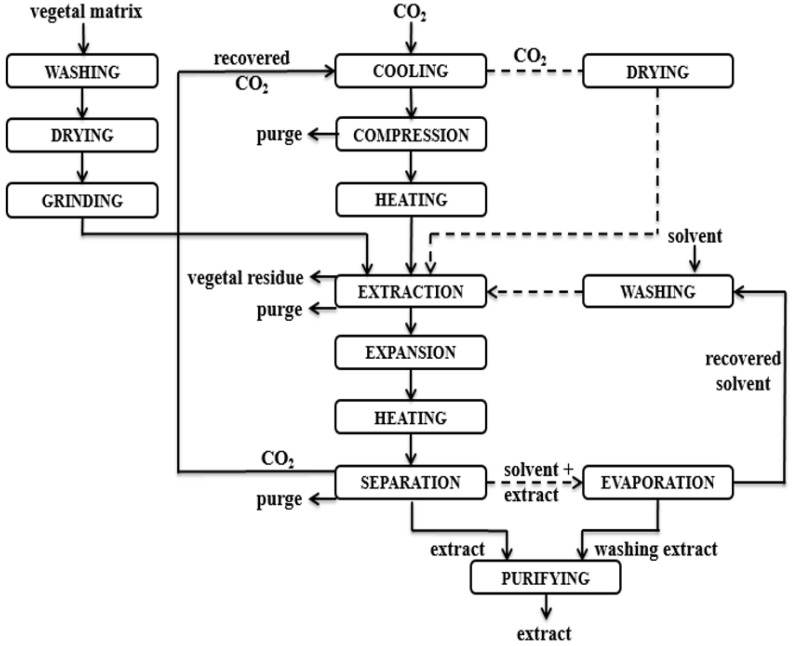
SFE flow diagram.

**Table 1 molecules-27-07737-t001:** The oil extraction yields of extracts obtained from *Lophanthus anisatus* by HD and BiAD.

Sample Type	Extraction Medium	Extraction Yield (g Extract/100 g Dried Plant ± SD)
Whole aerial plant, dried	HD	0.62 ± 0.020
Whole aerial plant, dried	BiAD	0.92 ± 0.015
Dried leaves	HD	0.75 ± 0.008
Dried leaves	BiAD	1.06 ± 0.005
Dried flowers	HD	1.22 ± 0.011
Dried flowers	BiAD	1.60 ± 0.0049

**Table 2 molecules-27-07737-t002:** Supercritical extraction samples and extracts of *Lophanthus anisatus*.

Sample Type	Dried Sample	Ground Sample	Extract	Extraction Yield (g Extract/100 g Sample ± SD)
Flowers	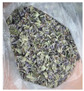	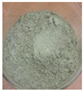	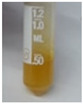	First stage 1.94 ± 0.030Second stage 0.57 ± 0.003
Leaves	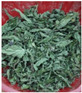	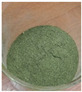	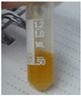	First stage 0.90 ± 0.010Second stage 1.14 ± 0.008
Whole plant	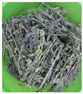	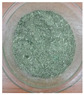	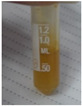	First stage 0.94 ± 0.010Second stage0.32 ± 0.007

**Table 3 molecules-27-07737-t003:** The extraction yield and total polyphenol concentrations determined by UV-VIS spectrometry.

Sample Name	Raw Material/Extraction Medium	Total Concentration Expressed as Gallic Acid Equivalent	
		[mg/L Extract]	[mg/g Dried Plant]
P1	Dried whole aerial plant/HD	1.7	4.25
P2	Dried whole aerial plant/BiAD	3.4	8.50
P3	Dried leaves/HD	3.8	9.50
P4	Dried leaves/BiAD	4.9	12.25
P5	Dried flowers/HD	3.2	8.00
P6	Dried flowers/BiAD	4.1	10.25
P7	Dried flowers/SFE stage 1	5.1	12.75
P8	Dried flowers/SFE stage 2	2.1	5.25
P9	Dried leaves/SFE stage 1	5.6	14.00
P10	Dried leaves/SFE stage 2	2.4	6.00
P11	Dried whole aerial plant/SFE stage 1	3.9	9.75
P12	Dried whole aerial plant/SFE stage 2	1.9	4.75

**Table 4 molecules-27-07737-t004:** Polyphenol concentrations in extracts obtained by HD, BiAD SFE, (mg/100 g dried plant).

Sample Name */Substance	P1	P2	P3	P4	P5	P6	P7	P8	P9	P10	P11	P12
Chlorogenic acid	5.25	16.35	28.80	33.63	22.05	31.50	33.90	16.70	41.70	21.30	36.30	19.65
Caffeic acid	2.85	11.01	8.10	11.85	1.95	3.45	7.20	4.40	14.20	12.75	11.00	8.00
Rosmarinic acid	120.60	220.50	204.00	309.00	191.10	312.00	277.50	138.00	279.00	128.40	227.00	130.80
Apigenin 7-glucoside	3.90	21.90	18.15	25.50	4.05	8.43	26.10	10.80	29.10	23.70	9.60	6.30
Ferulic acid	1.30	2.80	1.67	2.25	2.34	3.15	4.35	2.40	3.45	0.90	3.75	13.5

* P1–P12 meaning—the same as in Table 3.

**Table 5 molecules-27-07737-t005:** Chemical composition of EO obtained from *Lophanthus anisatus* by HD and BiAD.

Crt. No.	Component Denomination	P1	P2	P3	P4	P5	P6	
1	Chavicol, (p-Allylphenol)	1.04	1.27	14.22	-			
2	Estragole (methyl chavicol)	66.48	89.32	30.16	18.17	88.09	91.31	
3	Methoxy-eugenol	-	0.22	-	17.30	0.68	0.05	
4	2-Allyl-4-methoxyphenol)	1.22	-	-				
5	Limonene	5.24	5.40	-		8.01	7.07	
6	Methyl eugenol Eugenol methyl ether	9.77	0.22	-		0.68	0.10	
7	Caryophyllene	1.37	0.72	-				
9	(o-Allylphenol)	1.22	-	-				
10	(n-Octyl Acetate)	0.21	0.23	-		0.16		
11	Octanol	-	-	-		0.64	0.21	
12	Eugenol		0.04	13.96	17.96	0.37	0.17	
13	Benzaldehyde	5.2	0.55	2.44	10.16		0.12	
14	Pentanol	4.45	1.66	3.13	8.97		0.06	
15	Benzyl alcohol		-	2.44	3.10		0.09	
16	Phenyl ethyl alcohol		-	20.19	2.80		0.05	
17	Methyl jasmonate		-				0.51	
18	Ethyl lactate			4.53	2.63			
19	Cadinol α		-			0.10	0.08	

P1–P6 meaning—the same as in Table 3.

**Table 6 molecules-27-07737-t006:** Chemical composition of EO obtained from *Lophanthus anisatus* by SFE.

Crt. No.	Component Denomination	P7	P8	P9	P10	P11	P12
1	Estragole	88.89	91.41	79.5	80.17	92.37	93.04
2	Limonene	8.01	3.91	0.35		2.71	3.24
3	Caryophyllene	0.94	1.16			0.79	1.28
4	Germacrene	0.59	0.79			1.13	0.87
5	Octanol acetate	0.16	0.06			0.39	0.10
6	Elemene τ	0.20	0.14	0.48	0.37	0.18	0.26
7	Phellandrene	0.09	0.04	1.05		0.06	
8	Eugenol	0.37	0.28	4.57	4.57	0.57	0.06
9	Cadinol α	0.10	0.04	0.30	0.11	0.14	0.06
10	Phytol	0.15		0.73			0.10
11	Myrcene	0.07		0.22	0.10		
12	3 Octenone (Ethyl amyl ketone)	0.07		0.05	0.09		
13	Octenol 3 Ol (Vinyl amyl carbinol)	0.40	0.21	1.31			0.07
14	Terpineol	0.04				0.05	
15							
16	Methyl eugenol ether	0.68	1.74			1.23	0.65
17	Cadinene		0.04			0.22	0.06
18	Cubenol					0.03	0.04
19	2 Limonene			8.25	10.31	0.25	3.40
20	Phellandrene α			0.06	0.13		
21	Ethyl amyl ketone						
22	Octenol-1-ol, acetate (oct-1-enyl acetate)				0.21		
23	Caryophyllene β			1.06	1.42		
24	Germacrene D			0.95	1.02		
25	Cadinadiene				0.08		
26	Germacrene D-4-ol			0.07	0.10		
27	2 Eugenol		0.17	0.06	0.09		

P7–P12 meaning—the same as in Table 3.

**Table 7 molecules-27-07737-t007:** Antimicrobial activity of essential oils from *Lophanthus anisatus* and mint extract.

Tested Substances	Diameter of the Inhibition Zone (mm)		
	*Staphylococcus aureus*	*Escherichia coli*	*Pseudomonas aeruginosa*
EO *Lophanthus anisatus* flowers	18.0	0	0
EO *Lophanthus anisatus* leaves	9.0	0	0
EO *Lophanthus anisatus* whole plant	10.5	0	0
*Lophanthus anisatus* flowers (*infusion*)	0	12	7
*Lophanthus anisatus* flowers (*soak*)	0	0	9
Mint tea (infusion) (Maxi Pharma) (leaves and aerial part)	12	0	0
Mint tea (decoct) (Maxi Pharma) (leaves and aerial part)	10	0	7
Global interpretation	Intermediate	Resistant	Intermediate

**Table 8 molecules-27-07737-t008:** The antibiofilm effect.

Tested Substances	Antibiofilm Effect		
	*Staphylococcus aureus*	*Escherichia coli*	*Pseudomonas aeruginosa*
EO *Lophanthus anisatus* flowers	+	-	-
EO *Lophanthus anisatus* leaves	+	-	-
EO *Lophanthus anisatus* whole plant	+	-	-
Global interpretation	Active	Inactive	Inactive

**Table 9 molecules-27-07737-t009:** Extracts obtained from *Lophanthus anisatus* by HD and BiAD.

Sample Name	Raw Material	Extraction Method	Plant/Extraction Medium Ratio
P1	Whole aerial plant, dried	HD	2000 g/10 L distilled water
P2	Whole aerial plant, dried	BiAD	2000 g/10 L bio-alcohol solvent, 35 % alcoholic degrees
P3	Dried leaves	HD	2000 g/10 L distilled water
P4	Dried leaves	BiAD	2000 g/10 L bio-alcohol solvent, 35 % alcoholic degrees
P5	Dried flowers	HD	2000 g/10 L distilled water
P6	Dried flowers	BiAD	2000 g/10 L bio-alcohol solvent, 35 % alcoholic degrees

**Table 10 molecules-27-07737-t010:** SFE samples and operating conditions.

Sample Name	Raw Material	Extraction Medium	Extraction Conditions
P7	Dried flowers	Extraction stage 1	150 g dried flowers 8 h extraction time 13 kg/h CO_2_ flow rate 40 MPa pressure 40 °C temperature
P9	Dried leaves		
P11	Whole aerial plant, dried		
P8	Dried flowers	Extraction stage 2	The P7, P9, P11 sprayed with 10 g alcohol 4 h extraction time 13 kg/h CO_2_ flow rate 40 MPa pressure 40°C temperature
P10	Dried leaves		
P12	Whole aerial plant, dried		

**Table 11 molecules-27-07737-t011:** The working conditions for the mass spectroscopy determinations.

Element	Time (min)	Temperature (°C)
Column	0–10	40
	10–45	40–220
	45–55	220
Injector		200
Detector		235

## Data Availability

Not applicable.

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
