# Peer review of "Comparative Study of Useful Compounds Extracted from Lophanthus anisatus by Green Extraction"

_molecules, 2022, doi:10.3390/molecules27227737_

Round 1

Reviewer 1 Report

The present manuscript is dealing with the extraction of polyphenols and essential oils compounds from Lophanthus anisatus flowers, leaves, and whole plant through hydrodistilation, bio-solvent distillation, and super-critical fluids extraction. The manuscript presents important solutions for the environmentally friendly extraction of substances from valuable raw materials with the characteristics of the process. Such a study is extremely useful for reducing the number of tests under the conditions of organizing a production line. The conducted research can be considered reliable, taking into account the statistical processing of data and the evaluation of the composition of extracts with the proposed instrumental and analytical methods. In order to improve the general quality of the work, I suggest only some changes:

1) In Table 2 please correctly indicate the name of Rosmarinic acid. In Table 3 please correctly indicate the name of compound 27 (jasmine ketolactone?). Clarify the name of the 4th compound – “Alilmetoxifenol” (perhaps it means - 4-Allylmethoxyphenol? or 2-Allylmethoxyphenol?). Clarify the name of the compound 9 – Alilfenol (maybe it means – 2-Allylphenol?). The same applies to compound 1 – “Cavicol” (perhaps Chavicol is meant?). Same with compound 10 - “Octanol acetate” correct to - Octyl acetate.

2) In Table 4. please clarify the chemical names of terpineol and cadinene (alpha, delta, gamma?). 12th compound – “3 Octenone”, correct to - 3-Octenone. 13th compound – “Octenol 3 Ol”, correct to – Octen-3-ol. 22th compound – “Octenol-1-ol, acetate”, correct to - Octen-1-ol acetate. What is “Isoldene”? (compound 13), please write the chemical name in brackets.

3) Please indicate the value[g] +/- STD in the mass balance process scheme (Figure 3 a,b,c). Under Tables 2, 3 and 4, indicate the accuracy with which the obtained concentration are indicated.

4) The lines starting from 489-494 (In order to determine the composition…) describes the HPLC conditions (does not apply to GC), they should be moved to chapter 4.3.2, or placed under a separate subsection, as well as specify the conditions of the mobile phase and the gradient regime. The description of preparation of standard solutions and used concentration range is not indicated, please add it.

5) I do not see the need for Figure 6, because a qualitative explanation of the process is presented in Figure 7. I recommend remove the schematic diagram of supercritical fluid extraction plant.

6) From the study, the utility of using two-stage supercritical CO2 was not clear, considering that the described process resembles and also includes stages that are usually implemented in the combined extraction cycle with CO2 and cosolvent.

7) Why were plant samples dried at room temperature used for water extraction, but plants dried at 50 0C were used for CO2 extraction (line 405)? In the literature description, the effect of drying the plants on the yield of the extracted extract was reported. The question arises, why was this circumstance not respected?

8) The possibility of adding the obtained extracts to food as an antimicrobial component is mentioned in the discussion part (line 356). What would be the recommended amount of extract that could be added to food to obtain the desired antimicrobial effect without significantly altering the taste of the food?

Author Response

I upload an attachement.

Reviewer 2 Report

The reviewed paper represents a serious investigation on extraction of organic compounds from Agastache foeniculum plants by using different extraction techniques. The paper is well written and certainly under scope of the Molecules/MDPI. However, there are few comments. Major revision is recommended before publication.

General comments.

It is extremely difficult to follow the numerous data presented for many compounds in a big volume Introduction.

Firstly, there is excessive information on composition of essential oils obtained for the different types of plants in the genus Agastache including A. scrophulariifolia, A. Mexicana, A. Rugosa, Lophantus anisatum Benth. Clearly, the composition of EO for these plants may differ from in terms of both components and their concentrations from Agastache foeniculum. As experimental part is devoted only to the latter plant it would be useful to keep only information for Agastache foeniculum.

Secondly, there are many data on composition of Agastache foeniculum extracts obtained by different research groups. It would be useful to summarize them in the table for organic compounds, especially, for those, which are considered later in the experimental work. This will help to compare the types of compounds and their concentrations reported in the literature and obtained by the authors.  

More specific questions and comments.

Lines 136-138, two compounds 1,8-Cineol (Eucalyptol) and Globulol are listed as the most abundant components in EO of Agastache foeniculum. However, these compounds are not found by the authors in 28 components identified in their experimental work (see Table 3). Why?

The same question is about presence of Pulegone, Menthone, Caryophyllene and Isomenthone in the obtained EOs. For example, in non cited work of Chumakova et al. (full reference - Chumakova V.V., Popova O.I. Agastache foeniculum – A PERSPECTIVE SOURCE OF MEDICAL PRODUCTS. Pharmacy & Pharmacology. 2013; 1(1):39-43. (In Russ.) https://doi.org/10.19163/2307-9266-2013-1-1-39-43) were clearly identified as key components of EO obtained by hydrodistilation, which was used in the reviewed paper too. Actually, Pulegone (60.04%) and -Isomenthone (12.59%) are also mentioned in [14] as major components of EO (see line 80 of this manuscript).

A similar question is about presence of gallic acid, which is not listed in Table 2.

Line 78. It is not clear for which type of the Agastache the composition of EO is presented.

Author Response

I upload an attachment.

Reviewer 3 Report

The manuscript is well written, and the results are very and convincing. However, the discussion section needs to be improved. 

The disccusion is very poor. The article did not explains how (and why) their work agrees or disagrees with others or similar works. Besides, way forward based on findings not unanswered by your study, further research needed.

The conclusion of the study need call for actions or overview future research possibilities. 

Author Response

I upload an attachment.

Round 2

Reviewer 2 Report

The authors did a great job and answered all raised points. The paper looks stronger and can be recommended for the publication in Molecules/MDPI. 

Reviewer 3 Report

The manuscript has improved a lot after revision.